# Mycobacterial HelD is a nucleic acids-clearing factor for RNA polymerase

Tomáš Kouba [1,8 ✉], Tomáš Koval' [2,8], Petra Sudzinová [3,8], Jiří Pospíšil [3], Barbora Brezovská[3], Jarmila Hnilicová[3], Hana Šanderová [3], Martina Janoušková[3], Michaela Šiková [3], Petr Halada [3], Michal Sýkora [4], Ivan Barvík[5], Jiří Nováček[6], Mária Trundová[2], Jarmila Dušková[2], Tereza Skálová [2], URee Chon[7], Katsuhiko S. Murakami [7], Jan Dohnálek [2 ✉] & Libor Krásný [3 ✉]

RNA synthesis is central to life, and RNA polymerase (RNAP) depends on accessory factors for recovery from stalled states and adaptation to environmental changes. Here, we investigated the mechanism by which a helicase-like factor HelD recycles RNAP. We report a cryo-EM structure of a complex between the *Mycobacterium smegmatis* RNAP and HelD. The crescent-shaped HelD simultaneously penetrates deep into two RNAP channels that are responsible for nucleic acids binding and substrate delivery to the active site, thereby locking RNAP in an inactive state. We show that HelD prevents non-specific interactions between RNAP and DNA and dissociates stalled transcription elongation complexes. The liberated RNAP can either stay dormant, sequestered by HelD, or upon HelD release, restart transcription. Our results provide insights into the architecture and regulation of the highly medically-relevant mycobacterial transcription machinery and define HelD as a clearing factor that releases RNAP from nonfunctional complexes with nucleic acids.

[1] EMBL Grenoble, 71 Avenue des Martyrs, Grenoble, France. [2] Institute of Biotechnology of the Czech Academy of Sciences, Centre BIOCEV, Průmyslová 595, 252 50 Vestec, Czech Republic. [3] Institute of Microbiology of The Czech Academy of Sciences, Prague, Czech Republic. [4] Institute of Molecular Genetics of the Czech Academy of Sciences, Prague, Czech Republic. [5] Faculty of Mathematics and Physics, Institute of Physics, Charles University, Prague, Czech Republic. [6] CEITEC, Masaryk University, Brno, Czech Republic. [7] Department of Biochemistry and Molecular Biology, The Center for RNA Molecular Biology, Pennsylvania State University, University Park, PA 16802, USA. [8]These authors contributed equally: Tomáš Kouba, Tomáš Koval', Petra Sudzinová. ✉email: tkouba@embl.fr; dohnalek@ibt.cas.cz; krasny@biomed.cas.cz

A smoothly functioning transcription machinery is essential for maintaining the physiologically relevant levels of gene products and adequate changes in transcription are necessary for cell survival when the environment changes. In bacteria, transcription is executed by a single enzyme, DNA-dependent RNA polymerase (RNAP; composition of the core enzyme: $\alpha_2\beta\beta'\omega$[1]). The RNAP core is capable of transcription elongation and termination but not initiation. To initiate, a $\sigma$ factor is required to form a holoenzyme that recognizes specific DNA sequences, promoters[2]. RNAP holoenzymes can contain various $\sigma$ factors that allow interaction with diverse promoter sequences. The primary $\sigma$ factor is termed $\sigma^{70}$ in *E. coli* and $\sigma^A$ in most other species.

The two largest subunits, $\beta$ and $\beta'$, held together by the $\alpha$ dimer, form a crab claw-like structure (Fig. 1a), each subunit protruding into a pincer (the respective parts are called the $\beta$-protrusion and $\beta$-lobe [$\beta$-domain 1 and 2] and the $\beta'$-clamp). Subunits $\beta$ and $\beta'$ then form three channels[3]. The opening between the $\beta/\beta'$ pincers forms the primary channel where nucleic acids bind. The primary channel is separated by the bridge helix (BH; in $\beta'$) from the secondary channel, through which nucleoside triphosphates or other substrates[4] access the active site (AS) that is positioned at the junction of the two channels. Next to the BH, the trigger loop (TL, in $\beta'$) is found;

these two elements participate in the nucleotide addition cycle. Finally, the RNA exit channel lies on the opposite side of the RNAP core where nascent RNA passes between the base of the $\beta$-flap and the $\beta'$-lid. In the RNAP elongation complex (*Thermus thermophilus*, PDB ID 2O5J[5]), the downstream DNA (dwDNA) enters the complex through a cleft between the $\beta'$-clamp, $\beta'$-jaw, and $\beta$-lobe; the template strand then reaches the AS around the BH, and the DNA/RNA hybrid is held between $\beta'$-rudder, $\beta'$-lid, and $\beta$-protrusion.

Besides the RNAP subunits that are conserved in all bacteria, some species contain additional subunits, such as $\delta$ and $\varepsilon$ that are present in *Firmicutes*[6,7]. In addition, the regulation of the transcription machinery depends on concerted activities of RNAP and numerous transcription factors, such as RbpA in mycobacteria[8].

Another transcription factor is HelD[9], a protein similar to SF1 helicases[10] that associates with the RNAP core in the model Gram-positive bacterium *Bacillus subtilis* (*Bsu*) where it was shown to be involved in transcriptional recycling[11]. *Bsu* HelD binds and hydrolyzes ATP and this is accompanied by conformational changes in the protein as demonstrated by SAXS experiments[12]. The absence of HelD from *Bsu* cells results in a prolonged lag phase during outgrowth of stationary phase cells when diluted into fresh medium[11]. Overexpression of HelD then

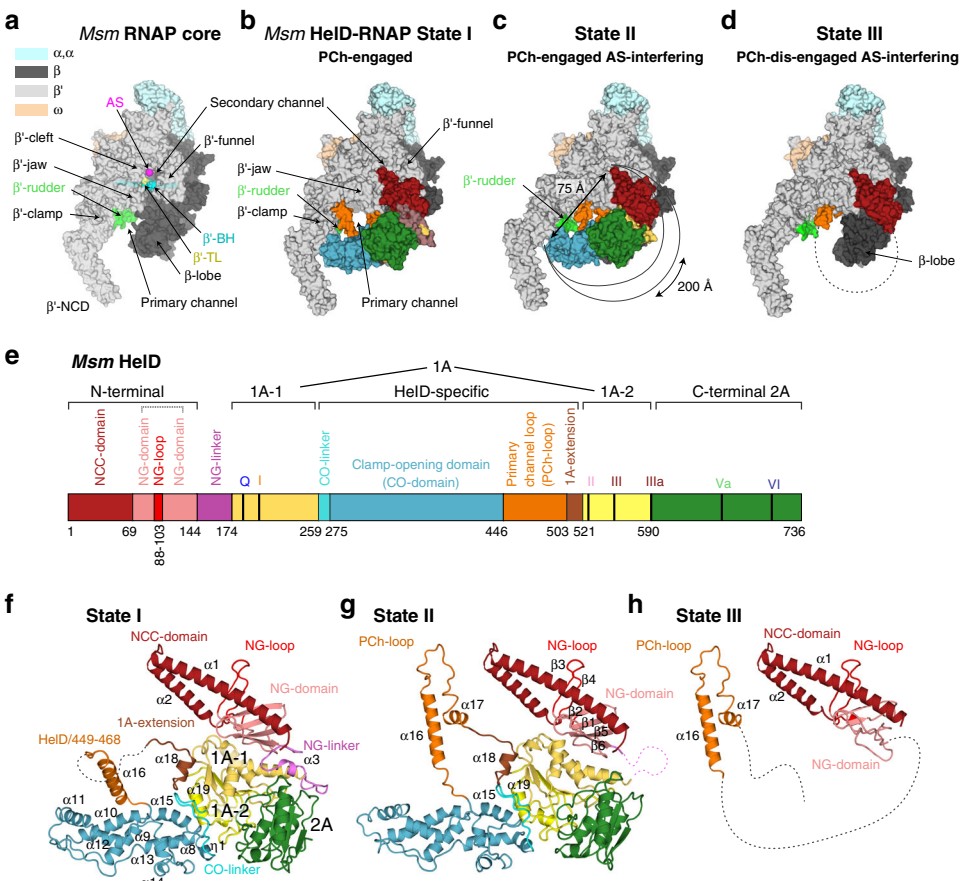

**Fig. 1 Cryo-EM structures of *Msm* HelD–RNAP complexes. a** Description of *Msm* RNAP core (PDB ID 6F6W) subunits and domains; RNAP subunits are color-coded according to the inset legend. **b–d** Atomic model surface representation of three identified *Msm* HelD–RNAP complexes: State I – PCh-engaged, State II – PCh-engaged AS-interfering, and State III – PCh-dis-engaged AS-interfering. When fully ordered in State I and II (**b, c**), the HelD protein (color-coded as in **e**) forms a crescent-like shape, ends of which protrude to the primary and secondary channels of the RNAP core. The partly ordered HelD protein in State III (**d**) vacates most of the RNAP primary channel. **e** Schematic linear representation of the domain structure of the HelD protein. The 1A domain (two shades of yellow) is split in aa sequence into two parts, separated by a large HelD-specific insertion (hues of blue and orange). The nucleotide-binding motifs are marked as vertical thick black lines. Aa numbering (*Msm*) is shown below. **f–h** Three states of HelD as observed in **b–d** color-coded according to the domain structure (**e**); secondary structure elements are marked as in Supplementary Fig. 7a.

accelerates spore formation[13]. However, the structure of HelD, its binding mode to RNAP, and mechanistic details of its function are unknown.

Here, we present structural data for HelD from *Mycobacterium smegmatis* (*Msm*) in complex with the RNAP core and provide insights into its function. We solved the 3D structures of three complexes of *Msm* RNAP and HelD by cryogenic electron microscopy (cryo-EM). The structures represent a so far unknown type of interaction between an RNAP and a protein. The structures suggested the possibility of simultaneous binding of HelD and σ[A] to RNAP, and by immunoprecipitation experiments we detected this transitional complex in the cell. Next, we provide biochemical evidence showing that in addition to being able to hydrolyze ATP, HelD can also hydrolyze GTP. Finally, we demonstrate that HelD can both prevent binding of the RNAP core to non-specific DNA and actively remove RNAP from stalled elongation complexes. Together, the results provide the basis for defining the role of HelD in the transcriptional cycle.

## Results

**Cryo-EM of *Msm* RNAP–HelD complex.** Our long-term attempts to crystalize *Bsu* HelD, RNAP core, or their complex failed; our cryo-EM experiments with the *Bsu* RNAP core were not successful; also, our recent SAXS-based data for the *Bsu* HelD–RNAP complex were not fully conclusive. However, in co-immunoprecipitation experiments with *Msm* RNAP, we identified MSMEG_2174, a potential homolog of *Bsu* HelD (Supplementary Fig. 1). We also solved the X-ray crystal structure of *Bsu* HelD C-terminal domain (CTD), which was then used as a guide for building the model of *Msm* HelD.

We reconstituted a complex of the *Msm* RNAP core and *Msm* HelD from purified recombinant proteins (Supplementary Fig. 2), and froze an isolated homogenous fraction of the complex on cryo-EM grids. We collected multiple preliminary cryo-EM data sets, which allowed us to optimize the cryo-EM conditions for high-resolution three-dimensional (3D) single-particle reconstructions (Supplementary Figs. 3–6). We identified two major 3D classes (State I and State II, Supplementary Fig. 4) at overall resolution ~3.1 Å (plus one subclass at ~3.6 Å), visualizing almost the complete structure of HelD bound to the RNAP core in two conformations (Fig. 1b, c, and Supplementary Movies 1, 2), and one minor class (State III; Supplementary Fig. 4), at ~3.5 Å, which delineates only two domains of HelD binding to the RNAP core (Fig. 1d and Supplementary Movie 3).

The structures of States I and II share the same overall fold of HelD, with a crescent-like shape (Fig. 1b, c). The main body of the crescent is sitting in between the β-lobe, the cleft/jaw, and the funnel/secondary channel of the β' subunit, burying about 774 and 2608 Å$^2$ in State I and 1490 and 3623 Å$^2$ in State II of the binding surface area of β and β' subunits, respectively[14]. One end of the crescent protrudes deep into the primary channel, and the other end into the secondary channel of the RNAP core. Indeed, to be able to reach both RNAP channels simultaneously, the HelD protein is markedly elongated, around 200 Å along the outer edge of the virtual crescent, and the two ends of the HelD protein are separated by ~75 Å (State II; Fig. 1c).

The HelD protein itself is divided into six structured domains (Fig. 1e–h), several of which possess unique, so far unknown folds. Interestingly, the 1A domain is composed of two parts (1A-1 and 1A-2) that are separated in the primary amino acid sequence by the intervening HelD-specific domain. According to the position of the HelD domains within the primary channel (PCh) and active site (AS), we name State I: PCh-engaged, State II: PCh-engaged and AS-interfering, and State III: PCh disengaged and AS-interfering (Fig. 1b–d).

**The HelD N-terminal domain inserts into the RNAP secondary channel.** The *Msm* N-terminal domain (HelD/1–144) forms an antiparallel α-helical coiled-coil (NCC) (HelD/1–69) followed by, and packed against the four-β-strand globular (NG) domain (HelD/70–144), which contains an additional prominent protruding loop (NG-loop, residues HelD/88–103; Figs. 1e–h and 2a, b). The overall N-terminal domain structure is analogous to the archetypal fold interacting with the secondary channel of RNAP present in transcription factors such as GreA or ppGpp cofactor DksA[15–17]. Indeed, the HelD N-terminal domain interacts tightly with the secondary channel, burying ~1790 Å$^2$ of the interaction surface, contributing largely to the HelD–RNAP interaction. Several specific hydrogen bonds and salt bridges (Supplementary Table 1) are formed between the N-terminal domain and the secondary channel, and particularly the NG-loop specifically recognizes the tip of the coiled-coil (CC) motif of the β'-funnel (Fig. 2a).

The topology of the *Msm* HelD NCC is conserved in comparison with other secondary channel-interacting transcription factors (Supplementary Fig. 8); however, in contrast to the known structures of such complexes, the *Msm* HelD NCC is shorter and its tip does not reach into the AS (Supplementary Fig. 8). Indeed, a large part of the NCC is extensively packed with the NG-domain into a common hydrophobic core, thereby preventing the NCC to bind further towards the AS. The HelD NCC tip is positioned at the level of the RNAP AS β' bridge helix (β'-BH), ~10–12 Å away from Mg$^{2+}$ metal A (MgA) of the AS, and as a result, it constitutes one wall of the secondary channel pore leading to the AS. The pore itself is ~11 Å wide (Fig. 2b) and this would still allow nucleoside triphosphate (NTP) passage into the AS. On the other hand, the NCC-domain restricts the conformational freedom and induces folding of the AS trigger loop (β'-TL, β'/1009–1028). This would likely interfere with the nucleotide addition cycle.

Another difference with respect to GreA family transcription factors is that the HelD NCC tip does not contain the conserved DXX(E/D)[18–20] motif (Supplementary Fig. 8), and it is, therefore, unlikely that the *Msm* HelD N-terminal domain possesses a Gre factor-like endonuclease activity.

**The NTPase unit of HelD is positioned in the vicinity of the downstream section of the primary channel.** The presented structure confirms our previous prediction[12] that HelD, similarly to SF1 helicases, RapA and UvrD, contains a conserved Rossmann fold 1A–2A heterodimer. Domain 1A is formed by two sub-domains 1A-1 and 1A-2 separated in amino acid sequence by the HelD-specific part (Fig. 1e). 1A-1 is connected with the N-terminal domain by the NG-linker (HelD/145–173), which orders only in State I. 1A-2 is then followed by 2A (Figs. 1e–g and 2c).

The 1A domain docks on the β-lobe where it induces small changes in domain orientation and conformation and it prolongs the wall of the downstream section of the primary channel along the axis of the virtual dwDNA (Fig. 2c). The 1A domain buries an area of 725 Å$^2$ of the interaction surface of the β-lobe, and the binding also involves the ordering of the β-turn β/209–212 and many hydrogen bonds and salt bridges (Supplementary Table 2). In addition, the extension of the 1A domain (HelD/504–521) is clamped in between the prominent β-turn β/184–187 of the β-lobe and the tip of the β'-jaw, further securing the 1A domain in its place (Fig. 2c).

The 1A–2A heterodimer establishes the canonical tertiary structure to form an NTP-binding pocket. Conserved residues of motifs Q, I, II, ~III, IIIa, Va, and VI are then likely involved in NTP binding[21] (Fig. 2d), while residues typical for DNA binding (in SF1 helicases) are missing. However, the base type specificity

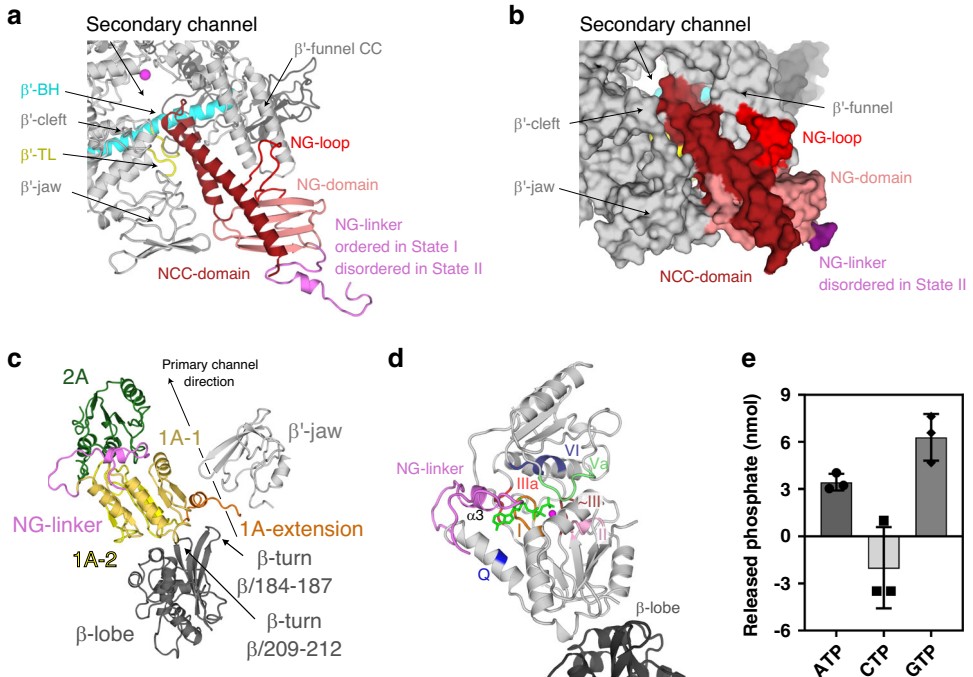

**Fig. 2 The HelD N-terminal domain inserts into the RNAP secondary channel; domains 1A–2A comprise the NTPase unit. a, b** Ribbon (State I) and surface (State II) representation of the HelD N-terminal domain interaction with the secondary channel of RNAP core (gray). The HelD coiled-coil domain (NCC-domain, firebrick) and the distinct loop (NG-loop, red) of the HelD globular domain (NG-domain, salmon) are inserted between β'-funnel, shelf, and jaw. The NCC-domain reaches only the boundary line of the β' bridge helix (β'-BH, cyan) and leaves a passageway to the RNAP core active site (MgA, magenta sphere). The HelD NCC also restricts the trigger loop (TL, yellow) movement. The linker (NG-linker, violet) connects the N-terminal domain with domain 1A-1. **c** The two *Msm* HelD Rossman fold domains (1A yellow and 2A green) form a canonical NTPase unit heterodimer with respect to structurally described SF1 helicases. Domain 1A tightly packs with β-lobe (dark gray) and its extension (brown) is clamped in between one β-turn (β/184–187) of β-lobe and the tip of the β' subunit jaw (light gray). **d** Model of ATP binding to the conserved nucleotide-binding site of motifs Q (blue), I (brown), II (pink), ~III (orange), IIIa (red), Va (pale green), and VI (deep blue). ATP (green) and Mg$^{2+}$ (magenta sphere) are added based on superposition with the ternary complex of UvrD (PDB ID 2IS4). **e** HelD exhibits ATPase and GTPase activities but does not hydrolyze CTP. The apparent negative value of CTP hydrolysis was caused by high background readings. The bars show mean values, the error bars indicate ±SD and the individual symbols represent values from three independent replicates. The data were analyzed and the graphics created with GraphPad Prism 7.02.

is not obvious from the structural data and, therefore, we measured nucleoside triphosphate hydrolysis activity of the isolated HelD protein. HelD showed strong hydrolysis activity of purine base nucleoside triphosphates but no activity towards a pyrimidine-containing counterpart (Fig. 2e and Supplementary Fig. 9f).

We also added ATP or non-hydrolyzable ATP analog to the HelD–RNAP complex, but we were not able to visualize any NTP-bound state by cryo-EM. Indeed, the orientations of conserved HelD/Tyr589 and Arg590 of motif IIIa, which are supposed to stack and coordinate the base and phosphate groups in the canonical ATP-bound state[22], are incompatible with NTP binding in the HelD NTP-free states (States I and II; Supplementary Fig. 9a). Notably, helix α3 of the ordered NG-linker in State I covers the putative NTP-binding pocket and partially obstructs the site entrance (Supplementary Fig. 9a). However, the entire linker can become disordered as seen in State II (Supplementary Fig. 10h), which is probably more compatible with NTP binding (see details below).

The superposition of HelD 1A–2A with similar structures of UvrD (PDB ID 2IS4) (Supplementary Fig. 9b, c), PcrA (PDB ID 3PJR), AdnA/B (PDB ID 6PPR), and RapA (PDB ID 6BOG) confirms that the Rossmann fold domains are packed in the canonical mutual orientation. However, unlike in bona fide SF1 helicases[21] where ssDNA is bound in the interface cleft of the dimer by conserved motifs Ia, Ic, IV, and V (Supplementary Fig. 9b, c), these motifs are not conserved in HelD. Instead, HelD contains proline-rich loops in place of these motifs and a large negatively charged surface patch in the equivalent areas (Supplementary Fig. 9d, e). Similarly, the ssDNA-binding motifs are not conserved in RapA, a functional homolog of HelD and a helicase-like protein involved in recycling of RNAP. RapA, however, binds differently to RNAP than HelD[23].

**The *Msm* HelD-specific domain is inserted into the downstream section of the RNAP primary channel.** The HelD-specific insertion domain is composed of the clamp-opening domain (CO-domain, HelD/261–447) and the primary channel loop (PCh-loop, HelD/448–503) (Figs. 1e and 3b–e). The CO-domain is an extended, mostly α-helical, and so far undescribed fold with no structural homologs (Supplementary Fig. 7b). On one side, the CO-domain packs against the 1A domain helix α19 and β-turn HelD/561–564. Additionally, the CO-1A interaction is stabilized by the CO-linker (HelD/259–275), which connects the two domains. In State I, the other side of the CO-domain, the CO-tip, butts against the three-stranded sheet of the β' non-conserved domain (β'-NCD) and an α-helix (β'/122–133) of the β'-clamp just preceding it (Fig. 3a, b, d). The only significant ordered part of the PCh-loop in State I, the protruding helix α16 (HelD/451–468), is erected against the β' three-stranded sheet (β'/1164–1210) and the α16 tip locks behind the helix-turn-helix motif β'/271–304 by HelD/Tyr466. Altogether, the α16

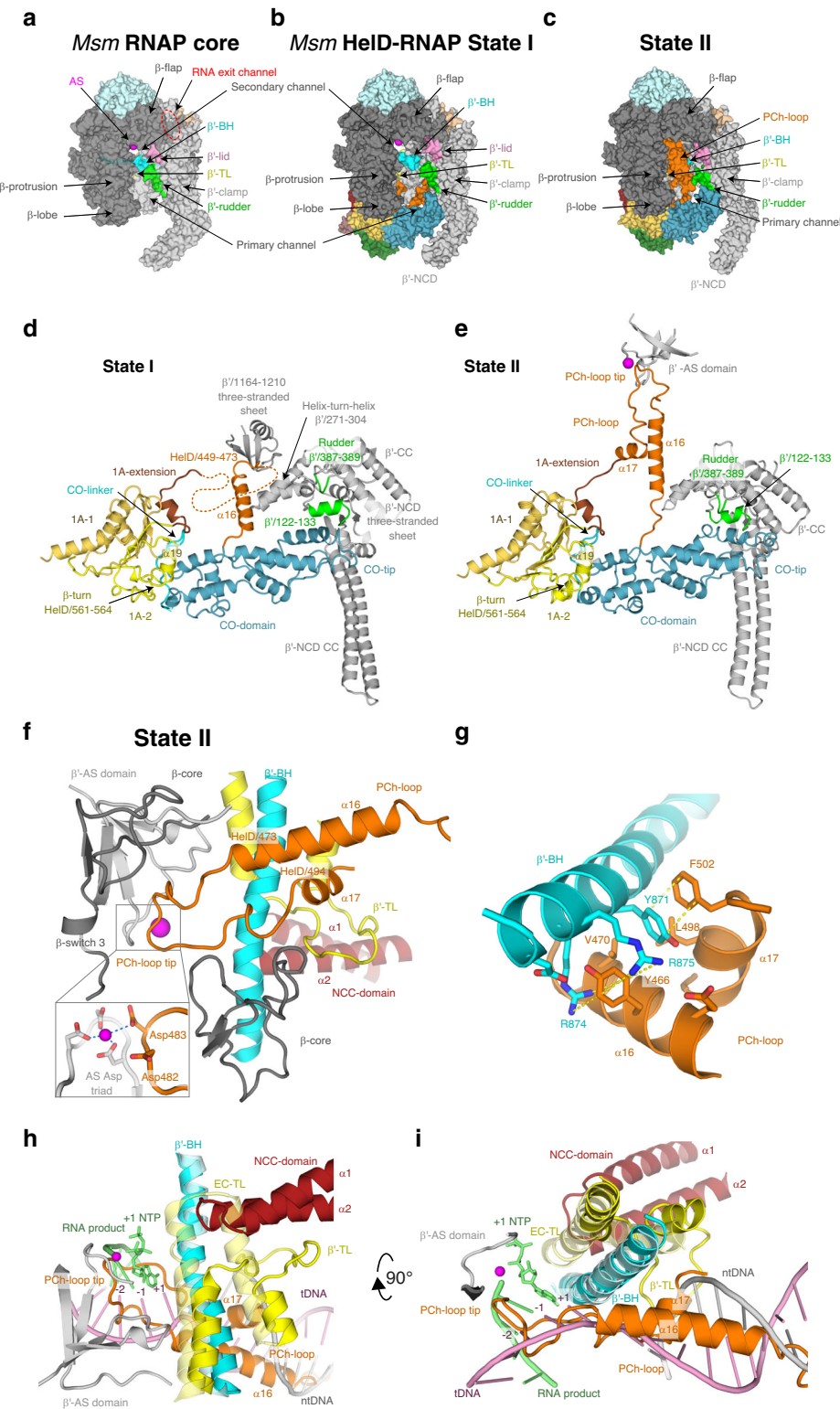

interaction with the β′-clamp might be helping the CO-domain insertion into the primary channel. In State II, the CO-domain fold alters and the PCh-loop completely refolds. The CO-domain tip shifts towards the β′ clamp coiled-coil domain (β′-CC) domain and reaches the peptide β′/387–389 of the rudder (Fig. 3c, e). The PCh-loop protruding helix α16 refolds (α16 register slightly shifts towards the C-terminus of HelD) and dis-engages with the β′ three-stranded sheet (β′/1164–1210), and the whole

PCh-loop orders towards the AS (see next section). Correspondingly, the two insertion modes of the CO-domain and PCh-loop into the primary channel force the β′-clamp domain to swing out into two distinct positions (see details below).

**The HelD PCh-loop is able to fold into the RNAP active site**. In the cryo-EM map of the AS-interfering State II, high-resolution density is present for the entire register of the PCh-loop, which is

**Fig. 3 The *Msm* HelD-specific domain interactions with the RNAP primary channel. a** Surface of the *Msm* RNAP core (PDB ID 6F6W), color-coded as in Fig. 1a with the description of individual domains and functional parts. **b**, **c** Surface representation of States I and II of the *Msm* HelD–RNAP complex with RNAP color-coding as in **a** and marked domain names; HelD color-coding as in Fig. 1e. **d**, **e** Ribbon representation of the HelD-specific domain inserting into the RNAP primary channel in State I (**d**) and State II (**e**). In State I (**d**), the clamp opening (CO, blue) HelD-specific domain is projected from the HelD 1A domain (yellow) towards the β'-clamp (gray). At one end, the CO is bonded to the 1A domain by the CO-linker (cyan), and stabilized by β-turn 561–563 and α19 (yellow). On the other end, the CO-domain tip abuts towards the β'-NCD three-stranded sheet. Concomitantly, the HelD helix α16 (part of peptide HelD/449–473, orange) butts against the β'/1164–1210 three-stranded sheet. The connection between α16 and the 1A-extension is disordered (dotted line). In State II (**e**), The CO interaction with the 1A domain remains similar to State I (**d**). The CO-domain tip, however, shifts towards the β'-rudder (green) and β'/122–133 α-helix. Concomitantly, the HelD PCh-loop (orange) folds towards the active site (MgA, magenta sphere) and folds back towards the 1A-extension (brick) and 1A domain. **f** The PCh-loop folds into the RNAP active site. The HelD loop 473–494 and the two adjacent α-helices (α16 and α17, orange) fold alongside the RNAP bridge helix (BH, cyan) towards the RNAP active site and HelD/Asp482 directly contacts the MgA (magenta sphere, details in the inset, coordination of MgA is marked with blue dotted lines). The RNAP trigger loop (TL, yellow) is restricted and folded between the HelD PCh-loop helix α17, the HelD NCC-domain (ruby), β'-BH, and the β-core domain (dark gray). **g** Detail of the β'-BH interaction with HelD α16 and α17. BH β'/Arg874 and Arg875 sandwich HelD/Tyr466, and β'/Tyr871 stacks on HelD/Phe502. The stacking interactions are marked with yellow dotted lines. **h**, **i** The HelD PCh-loop binding in the active site chamber is mutually exclusive with the presence of the transcription bubble. Two perpendicular views of superposition of the *Tt* RNAP elongation complex (PDB ID 2O5J, pale colors) and HelD State II (solid colors) are shown. The folded TL in pre-translocated EC would sterically clash with the HelD NCC-domain. The HelD PCh-loop tip would sterically clash with RNA/DNA hybrid at positions +1 to −2, and the HelD α16 and α17 helices would clash with downstream DNA duplex. Color code as in **f**, template DNA in pink, non-template DNA in gray, product RNA and incoming NTP at position +1 in green.

folded in the AS cavity of RNAP (Fig. 3c, f, g and Supplementary Fig. 5c). The folding of the PCh-loop in between the walls of the AS chamber is also compatible with the regular open form of the RNAP core as observed in State III.

In comparison to State I, in State II the protruding helix α16 refolds, the helix register shifts to residues 455–472, and together with a newly folded helix α17 (HelD/495–500) they tightly pack with the second half of the β'-BH (Fig. 3g). In detail, BH β'/Arg874 and Arg875 sandwich α16 HelD/Tyr466 and, cooperatively, BH β'/Tyr871 stacks on HelD/Phe502 and is inserted into a hydrophobic pocket formed by HelD/Tyr466, Ala467, Val470, and Leu498. The rest of the PCh-loop (HelD/473–494) specifically wedges into the AS cavity (Supplementary Table 3), towards the AS aspartate triad and MgA. Notably, there are four acidic residues (482–DDED-485) at the very tip of the PCh-loop and the HelD/481–483 peptide folds along the AS β-strand β'/537–544, such that HelD/Asp483 is in contact with MgA and HelD/Asp482 in its near proximity (Fig. 3f and Supplementary Fig. 5c). HelD/Asp482 interacts with β'/Arg500, HelD/Glu484 stabilizes the loop in the active site by interaction with β/His1026, and HelD/Asp485 contributes to the AS-interfering loop stability by a salt bridge with the side chain of HelD/Arg477. Two other motifs support the formation of the PCh-loop structure in the RNAP AS—a small hydrophobic core formed by the HelD/Val475, Leu480, and Leu488 side chains and an intra-chain ion-pair HelD/Arg477–Asp491, with HelD/Arg477 leaning against β/Pro483.

As a result of the PCh-loop folding into the primary channel and HelD NCC folding in the secondary channel, the NCC tip and the tip of the PCh-loop are brought close together (the shortest distance between the two tips is about 17 Å). This also restricts the trigger loop, which is, therefore, partially folded in the space between the BH, HelD α2 and α17, the peptide between α17 and α18, and the peptide of β/Ile182-Glu187. In summary, the PCh-loop seems to interfere with the AS cavity so that it is not compatible with the NTP addition cycle. Moreover, the super-position with the structure of *Thermus thermophilus* (*Tt*) RNAP EC (PDB ID 2O5J; Fig. 3h, i) suggests that the whole PCh-loop would be in steric clash with the dwDNA duplex and the RNA/DNA hybrid in the AS as far as position −2. A parallel can be drawn between the presence of the PCh-loop in State II and the so-called DNA-mimicking loop of PolI[24], which also occupies the AS chamber and the surroundings of the AS and is sterically incompatible with the presence of the DNA transcription bubble in RNAP.

**Global domain changes of RNAP upon HelD binding.** Super-position based on the β-core region (β/430–738) of the *Msm* RNAP core (PDB ID 6F6W), elongation complex (EC, model based on PDB ID 2O5J) and States I–III enables analyses of global differences of the three observed structural states (Supplementary Fig. 10). The interaction of the HelD N-terminal domain with the secondary channel and its influence on the rest of the complex remains very similar in all the States. This interaction thus might be the initial one through which HelD starts its association with RNAP. Furthermore, this interaction seems sufficient to alter the position of the β'-jaw/cleft and β-lobe (Supplementary Fig. 10g) which may weaken interaction with dwDNA, reminiscent of TraR (a distant DksA homolog) binding to *E. coli* RNAP[25].

The main change between the States is the interplay between the refolding of the PCh-loop and the CO-domain position in the primary channel. In State III, solely the PCh-loop's tight contact with the AS stabilizes a very open form of RNAP (Supplementary Fig. 10a, b, f), ~33 Å at the narrowest point of the primary channel (measured by the distance of the Cα atoms of β/Lys273 and β'/Lys123), comparable to the structures of two previously identified conformations of very open forms of *Msm* RNAP core and holoenzyme, termed Core2 and Holo2 (32.2 and 33.6 Å, respectively)[1]. In State I, the PCh-loop's interaction with β' helix-turn-helix and three-stranded sheet, and the CO-domain insertion into the primary channel make the opening of the RNAP clamp (~35 Å; Supplementary Fig. 10a, b) slightly wider than the already widely open forms of the Lipiarmycin-[26] (PDB ID 6FBV) and Fidaxomicin-locked[27] (PDB ID 6C06) RNAPs (34.2 and 33.6 Å, respectively)[1]. In State II (Supplementary Fig. 10e), while the CO-domain still inserted, the PCh-loop abolishes the β' contact and folds in the AS instead, and this forces the β'-clamp (β'/1–406) to rotate with respect to the remaining parts of the complex so that the β'-NCD CC tip opens further away from the juxtaposed β-lobe but at the same time the β'-rudder, β'-CC, and adjacent secondary elements move about 11 Å closer to the tip of the HelD CO-domain. The RNAP clamp is, therefore, splayed by 45 Å (Supplementary Fig. 10b). This clamp opening together with the tight interaction of the PCh-loop with the AS is not compatible with nucleic acid binding.

The next major differences are the β-lobe and CO-domain adjustments upon change of the 1A–2A heterodimer (Supplementary Fig. 10e). The mutual orientation of 1A and 2A domains between States I and II is almost preserved, although with much poorer density for 2A in State II. This most likely stems from the more pronounced mobility of 2A, possibly linked with the lack of stabilization by the unfolded NG-linker in State II. The 2A relaxation allows movement of 1A in respect to the N-terminal domain (~3° difference measured by HelD α1 and α5) and a concomitant shift of both the β-lobe and CO-domain (Supplementary Fig. 10e). In detail, this global change is accompanied by a shift and changes in the secondary structure of HelD/230–252 within the 1A domain (largest shift about 9.3 Å for Val245). Helix α6 is extended and helix α7 is formed in State II (Supplementary Fig. 7a) and 1A-extension shifted. State I interactions between α6 and the NTPase site, and α6 and the NG-linker that are NTP-binding prohibitive, are broken in State II and the NTPase site of HelD becomes wide open (NTP-binding permissive; Supplementary Fig. 10h). Although this change makes the NTPase site accessible for NTPs, additional conformational changes are still required for NTP accommodation.

Finally, HelD binding in States I and II also leads to the opening of the RNA exit channel between the β-flap and β′-lid and β′-Zn-finger by about 15 and 21 Å, respectively (Supplementary Fig. 10c, d). State III keeps the channel still rather open by about 12 Å. This is expected to contribute to RNA release.

**HelD clears the RNAP primary channel**. The position of the HelD CO-domain in the primary channel of RNAP suggests that HelD may prevent non-specific interactions between the RNAP core and DNA. To test this, we performed an electrophoretic mobility shift assay (EMSA) with RNAP and a fragment of mycobacterial DNA in the presence/absence of HelD. Figure 4a–c shows that HelD significantly abolishes the non-specific binding of the RNAP core to DNA.

Moreover, we speculated that HelD might not only prevent DNA binding but also actively disassemble stalled ECs. Stalled ECs (due to, e.g., damaged DNA) are obstacles for both the coupled transcription–translation machinery[28,29] and also for replication[30], with potentially deleterious consequences if not removed. To test the ability of HelD to rescue stalled RNAP, we assembled ECs with the RNAP core on a DNA–RNA scaffold and challenged them with HelD in the presence/absence of NTPs (Fig. 4d). HelD then, relative to mock treatment, was able to disassemble stalled ECs (Fig. 4e). This process, interestingly, appeared to be independent of ATP or GTP.

**HelD, σ^A, and RbpA can simultaneously bind RNAP core**. Analysis of States I–III suggested the possibility of simultaneous binding of HelD, σ^A, and RbpA to RNAP. Modeling of hypothetical complexes of RNAP–HelD with σ^A and RbpA then confirmed that relatively small changes in conformations of these proteins could allow their simultaneous binding to RNAP in States I–III (Supplementary Fig. 11 and "Discussion"). Therefore, we tested experimentally whether the HelD–RNAP complex is compatible with the presence of other factors. Indeed, immunoprecipitation (IP) and western blot experiments with FLAG-tagged Msm RNAP revealed the presence of HelD and σ^A (Fig. 4f, g); FLAG-tagged Msm σ^A pulled down the RNAP core and HelD; FLAG-tagged HelD pulled down the RNAP core and σ^A. These results suggested but not proved that HelD, σ^A, and RNAP are together in one complex. Alternatively, HelD and σ^A could bind each other independently of RNAP. To decide between the two possibilities, we first pulled down FLAG-tagged HelD and associated proteins and from this mixture, we subsequently pulled

down σ^A (with an antibody against σ^A) and associated proteins. Supplementary Fig. 12 shows the presence of HelD and RNAP in the second pull-down, demonstrating that all these proteins (RNAP, σ^A, HelD) can coexist in one complex. Additionally, RbpA, albeit in low amounts, was also present in the HelD-immunoprecipitated complex and RbpA-FLAG pulled down RNAP with σ^A and HelD (Supplementary Fig. 13). We then confirmed the interactions between the RNAP core, σ^A, RbpA, and HelD by in vitro EMSA (Fig. 4h).

**Discussion**
This study describes a structurally unique complex between Msm RNAP and the HelD protein, defines its DNA-clearing activity, and outlines its role in transcription.

**Comparison of *M. smegmatis* and *B. subtilis* HelD**. Previous biochemical studies used HelD from Bsu, which is only 21% identical with the Msm homolog. Selected sequence homologs of Msm HelD are shown in Supplementary Fig. 14, revealing two main differences between Msm (Actinobacteria) and Bsu (Firmicutes). The first marked difference is the absence of ~30 aa from the N-terminal NCC-domain region in Msm HelD. This is consistent with the Bsu HelD NCC-domain protruding much deeper into the RNAP secondary channel and even overlapping with the AS[31,32]. The other difference is in the HelD-specific region where Bsu HelD completely lacks the PCh-loop. On the other hand, the organization of the 1A-1 and 1A-2 split followed by the complete 2A domain is maintained (Fig. 1e, f, g).

Interestingly, Msm HelD, σ^A, and RbpA can co-occur on RNAP (Fig. 4h and Supplementary Figs. 11–13) and we infer that the RNAP–σ^A–RbpA–HelD complex thus likely represents one of the possible transitional states in the transcriptional cycle. This differs from Bsu where simultaneous HelD and σ^A binding has not been detected[11]. Regardless of the exact mutual positions of σ^A, RbpA, and HelD, RNAP must subsequently assume a conformation that is compatible with promoter DNA binding and transcription initiation.

**Model of the HelD role in transcription**. Based on the structural and functional data we propose a role for Msm HelD in transcription (a model is shown in Fig. 5). We envisage that upon transcription termination when RNAP fails to dissociate from nucleic acids[33], or in the event of stalled elongation, Msm HelD first interacts with RNAP by its N-terminal domain, likely competing for binding to the secondary channel with GreA-like factors. This initial HelD binding induces changes in β-lobe and β′-jaw/cleft (Supplementary Fig. 10g), possibly leading to destabilization of dwDNA in the primary channel. The trigger loop is conformationally locked. Subsequently, the CO-domain and PCh-loop approach the primary channel. The PCh-loop, which is probably flexible in the RNAP-unbound state, folds partially upon binding RNAP (captured in State I), and then it penetrates deep into the primary channel, fully folds, and binds to the AS (captured in States II and III). The CO-domain interactions with β′-clamp then secure the primary channel wide open (Supplementary Fig. 10a, b). At the same time, the RNA exit channel dilates (Supplementary Fig. 10c, d). All these processes lead to the release of any contents of the AS (compare states within Supplementary Fig. 10a).

We note that neither HelD loading onto RNAP nor RNAP clamp opening nor EC disassembly is dependent on NTP hydrolysis. Energy from NTP hydrolysis is probably required to release HelD from its tight contact with RNAP. Free energy corresponding to ATP hydrolysis under physiological conditions in cells is around −50 kJ/mol[34]. This is comparable to the

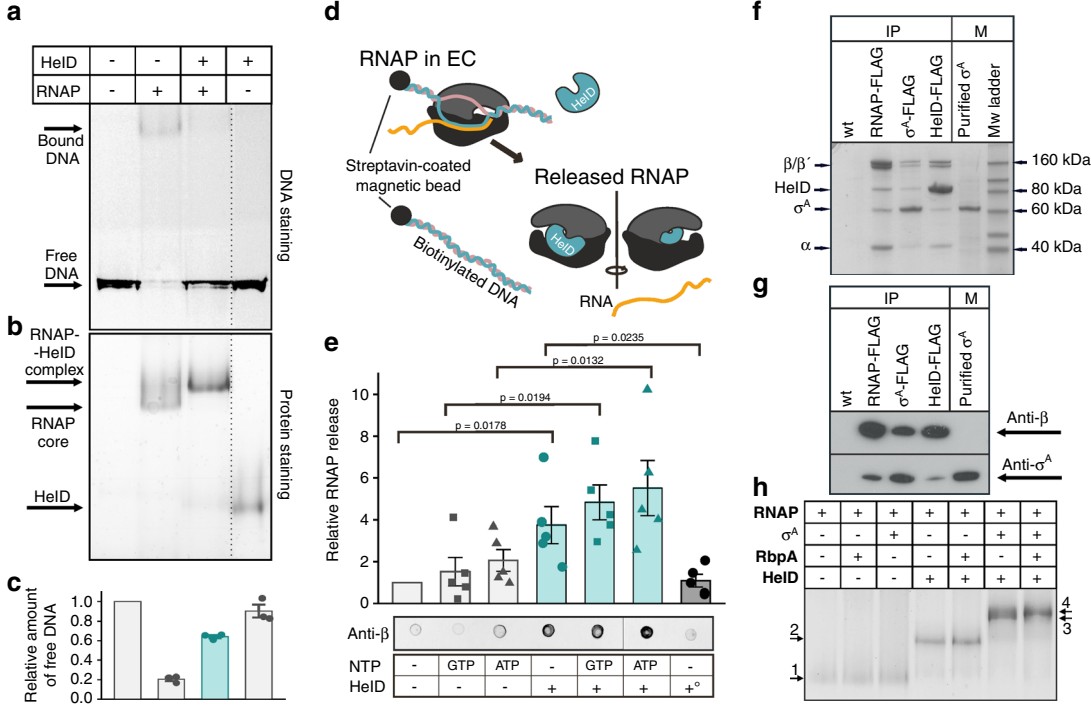

**Fig. 4 Binding of *Msm* HelD to RNAP and its effects on DNA–RNAP interactions. a** DNA binding to RNAP - EMSA - binding of 300 bp DNA to the *Msm* RNAP core and the effect of HelD. **b** The same gel as above but stained for proteins. The dotted line shows where the gel was electronically assembled. **c** Quantitation of EMSA – the bars here (the amount of unshifted DNA) and in **e** are mean values from at least three independent experiments, the error bars show ±SD, the individual symbols show values of individual independent replicates. The leftmost bars were set as 1 and the other values within each graph were normalized relative to this bar. The turquoise bars here and in **e** indicate the addition of HelD. **d** EC disassembly - scheme: ECs were assembled on DNA:RNA scaffolds and challenged with HelD and/or NTPs. RNAP released into buffer was quantitated by western dot blots. **e** Quantitation of EC disassembly from five independent experiments. Representative primary data are shown below the graph. Presence/absence of individual components is indicated. +° indicates heat-inactivated HelD. The statistical significance in **e** for the indicated combinations was $p < 0.05$ (one-sided Student's *t*-test; exact *p*-values are written in the graph). **f** Representative SDS-PAGE of immunoprecipitations of *Msm* RNAP ($\beta$), $\sigma^A$, and HelD. All proteins were FLAG fusions, the antibody was anti-FLAG. Wt, a strain without any FLAG fusion. The identity of the bands was confirmed by mass spectrometry. IP, immunoprecipitation; M, markers. The experiment (biological replicates) was performed 3× with the same result. **g** Representative western blot of IPs of FLAG-tagged *Msm* RNAP ($\beta$), $\sigma^A$, and HelD. Antibodies against RNAP $\beta$ and $\sigma^A$ were used to detect the presence of proteins in complexes. M, marker – purified $\sigma^A$. The experiment was performed twice with the same result. **h** In vitro protein interactions - EMSA. Proteins were detected by Simply blue SafeStain. In all cases, RNAP was first reconstituted with HelD and then with RbpA and/or $\sigma^A$. A small, but a reproducible shift was observed after the addition of both RbpA and $\sigma^A$ to RNAP–HelD, indicating the presence of all proteins in one complex. Numbered arrows indicate complexes with different protein composition (determined by mass spectrometry). In some cases, complexes with different protein compositions displayed the same migration in the gel: 1. RNAP, RNAP-RbpA; RNAP-$\sigma^A$; 2. RNAP–HelD, RNAP–HelD-RbpA; 3. RNAP–HelD-$\sigma^A$; 4. RNAP–HelD-$\sigma^A$-RbpA. The experiment (biological replicates) was performed 3× with the same result.

estimated desolvation energy of the HelD–RNAP core interaction of $-33.5$ kJ/mol ($\Delta^i G$) for State I and $-57$ kJ/mol for State II. However, States I and II are not fully compatible with canonical NTP binding in the HelD NTPase unit. It remains to be answered which structural changes are required to actually enable NTP binding and hydrolysis.

To summarize, HelD clears RNAP of nucleic acids; this likely happens in non-functional (e.g., stalled) transcription complexes or post-termination. This may contribute to the smooth functioning of the transcription machinery. Furthermore, it is conceivable that HelD may also function similarly to 6S RNA[35] or Ms1[36], which keep RNAP in an inactive state under growth-unfavorable conditions. This stored RNAP then accelerates the restart of gene expression when conditions improve.

Finally, the RNAP-inactivating ability of HelD might be utilized in the development of specific antibacterial compounds that would stabilize the non-productive HelD–RNAP complex, shifting the equilibrium of RNAP states towards effective transcription inhibition, as seen, e.g., in the action of Fidaxomicin towards *M. tuberculosis* RNAP in complex with RbpA[27].

## Methods

**Bacterial strains, plasmids, and oligonucleotides**. Bacterial strains and plasmids are listed in Supplementary Table 4. DNA oligonucleotides are listed in Supplementary Table 5.

**Strain construction—$\sigma^A$ and RbpA**. $\sigma^A$ (*MSMEG_2758*) and *rbpA* (*MSMEG_3858*) genes were amplified from genomic DNA by PCR with Phusion High-Fidelity DNA Polymerase (NEB) with primers #1155 + #1156 ($\sigma^A$) and #1182 + #1183 (RbpA) and *Msm* chromosomal DNA as the template, cloned into pET22b via *NdeI/XhoI* restriction sites and verified by sequencing. The resulting plasmids were transformed into expression *Eco* BL21(DE3) strain resulting in strains LK1740 ($\sigma^A$) and LK1254 (RbpA).

**Strain construction—HelD**. Plasmid encoding the N-terminally His-tagged *Msm* HelD protein was prepared by the GeneArt® Plasmid Construction Service (ThermoFisher). Gene construct for HelD expression was designed by codon-optimized back translation of gene MSMEG_2174 from *Msm* (strain ATCC

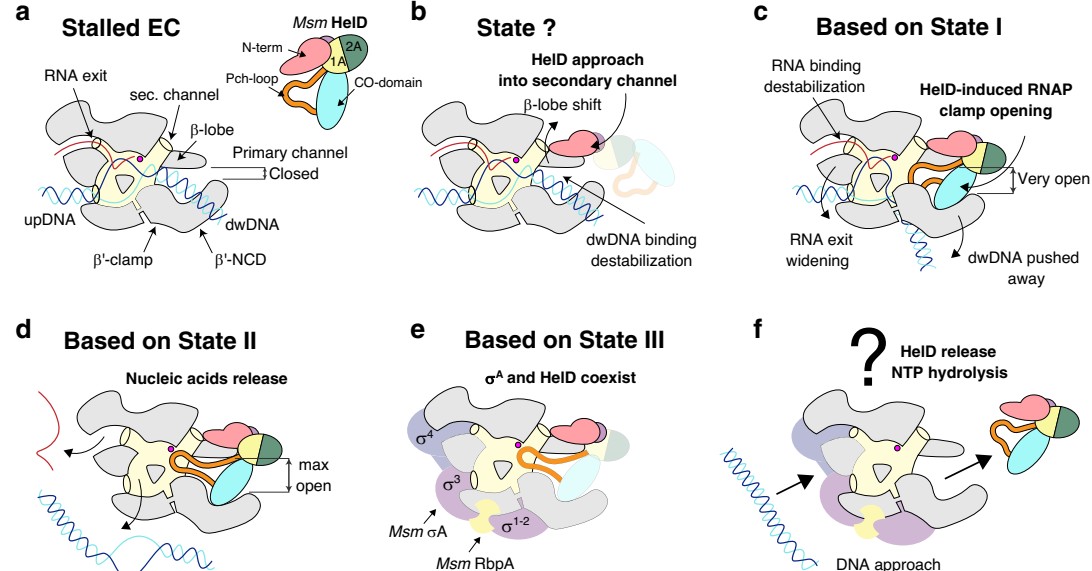

**Fig. 5 A model of the HelD functioning in RNAP recycling. a** When EC stalls, it needs to be disassembled. **b** The HelD N-terminal domain (pink) first approaches the RNAP secondary channel and then induces changes in RNAP likely destabilizing the RNAP-dwDNA interaction. **c** Subsequent interactions of the HelD PCh-loop (orange) and the whole HelD-specific domain (cyan) in the RNAP primary channel open the RNAP cleft, widen the RNA exit channel and mechanically interfere with dwDNA. **d** An even broader cleft/RNA exit opening together with the PCh-loop intervening deep in the AS (MgA, magenta sphere) displace dwDNA and the RNA/DNA hybrid from the active site cavity. **e** The HelD–RNAP nucleic acid-free complex binds σ^A factor and RbpA, and all factors can bind RNAP core simultaneously. **f** The complex binds to DNA promoter via the σ^A factor with a concomitant displacement of HelD from RNAP by an unknown mechanism, possibly dependent on NTP hydrolysis by HelD and a new round of σ^A-dependent transcription cycle can initiate.

700084/mc2 155) with cleavage site for TEV protease placed at the 5′ end. This synthesized gene was cloned into the Champion™ pET302/NT-His expression vector (Thermofisher) via *EcoRI* and *XhoI* restriction sites. The resulting protein thus has 6xHis tag at its N terminus, which is cleavable by TEV protease (protein construct starts with sequence MHHHHHHVNSLEENLYFQG followed by the second amino acid of gene *MSMEG_2174*).

**Strain construction—HelD-FLAG, σ^A-FLAG, and RbpA-FLAG**. The genes coding for the HelD-FLAG, σ^A-FLAG, and RbpA-FLAG proteins were amplified by PCR using Q5® High-Fidelity DNA Polymerase (NEB) with primers #3130 + #3131 (HelD), #2339 + #2340 (σ^A) and #2894 + #3093 (RbpA) and *Msm* chromosomal DNA as the template. The C-terminal 1× FLAG-tags (DYKDDDDK) were encoded within the reverse PCR primers for all genes. Subsequently, the genes were inserted into integrative plasmid pTetInt[37] via *NdeI*/*Hind*III restriction sites. The constructs were verified by sequencing. The resulting plasmids were transformed into *Msm* mc2 155 (wt, LK865) cells by electroporation resulting in strains LK2590 (HelD-FLAG), LK2073 (σ^A-FLAG), and LK2541 (RbpA-FLAG).

**Growth conditions**. *Msm* strains—mc2 155 (wt, LK865), σ^A-FLAG (LK2373), RNAP-FLAG (LK1468), HelD-FLAG (LK2590), and RbpA-FLAG (LK2541) were grown at 37 °C in Middlebrook 7H9 medium with 0.2% glycerol and 0.05% Tween 80 and harvested in exponential phase (OD_600 ~ 0.5; 6 h of cultivation) or early stationary phase (OD_600 ~ 2.5–3.0, 24 h of cultivation) unless stated otherwise. When required, media were supplemented with kanamycin (20 μg/ml). Expression of HelD-FLAG in exponential phase was induced by anhydrotetracycline (1 ng/ml) at 3 h of cultivation. The cells were then grown for an additional 3 h. Expressions of σ^A-FLAG, RbpA-FLAG, and HelD-FLAG in the stationary phase were induced by anhydrotetracycline (10 ng/ml) at 8 h of cultivation. The cells were then cultivated for an additional 16 h.

**Msm RNAP core purification for cryo-EM**. *Eco* strain BL21(DE3) was transformed with pRMS4 (*kanR*) plasmid derivative encoding *Msm* subunits ω, α, and β–β′ fusion with C-terminal His8 tag in one operon from the T7 promoter. Expression cultures were incubated at 37 °C and shaken at 250 rpm until OD_600 ~ 0.8; expression was induced with 500 μM isopropyl β-D-thiogalactoside (IPTG) at 17 °C for 16 h. Cells were lysed using sonication by Sonic Dismembrator Model 705 (Fisher Scientific) in a lysis buffer containing 50 mM NaH_2PO_4/Na_2HPO_4 pH 8 (4 °C), 300 mM NaCl, 2.5 mM MgCl_2, 30 mM imidazole, 5 mM β-mercaptoethanol, EDTA-free protease inhibitor cocktail (Roche), RNase A (Sigma), DNase I (Sigma), and Lysozyme (Sigma). Clarified lysate was loaded onto a HisTrap FF Crude column (GE Health-care) and proteins were eluted with a linear gradient of imidazole to the final

concentration of 400 mM over 20 column volumes. The *Msm* RNAP core elution fractions were pooled and dialyzed to 20 mM Tris–HCl pH 8 (4 °C), 1 M NaCl, 5% (v/v) glycerol and 4 mM dithiothreitol (DTT) for 20 h. The protein was further polished on XK 26/70 Superose 6 pg column (GE Healthcare) equilibrated in 20 mM Tris–HCl pH 8 (4 °C), 300 mM NaCl, 5% (v/v) glycerol, and 4 mM DTT. The *Msm* RNAP core final fractions were eluted at 6 μM concentration, aliquoted, flash-frozen in liquid nitrogen, and then stored at −80 °C.

**Msm HelD protein purification for cryo-EM**. *Eco* strain Lemo 21 (DE3) was transformed with pET302/NT-His (*cmlR* and *ampR*) plasmid derivative encoding the *Msm* HelD protein fusion with N-terminal 6×His tag under the control of the T7 promoter. Expression cultures were incubated at 37 °C and shaken at 250 rpm until OD_600 ~0.8; expression was induced with 500 μM IPTG at 17 °C for 16 h. Cells were lysed using sonication by Sonic Dismembrator Model 705 (Fisher Scientific) in a lysis buffer containing 50 mM Tris–HCl pH 7.5 (4 °C), 400 mM NaCl, 30 mM imidazole, 0.2% Tween20, 2 mM β-mercaptoethanol, EDTA-free protease inhibitor cocktail (Roche), RNase A (Sigma), DNase I (Sigma), and Lysozyme (Sigma). Clarified lysate was loaded onto a HisTrap FF Crude column (GE Healthcare) and proteins were eluted with a linear gradient of imidazole to the final concentration of 400 mM over 20 column volumes. Fractions containing HelD protein were pooled and dialyzed for 20 h against the dialysis buffer containing 20 mM Tris–HCl, pH 7.5 (4 °C), 500 mM NaCl, 1 mM DTT together with TEV protease at a TEV protease:HelD ratio 1:20.

The protein was then concentrated to ~15 A_280 units and further purified using size-exclusion chromatography using a Superdex 75 column (GE Healthcare) equilibrated in 20 mM Tris–HCl, pH 7.5 (4 °C), 200 mM NaCl and 1 mM DTT. The HelD protein was eluted at ~160 μM concentration, aliquoted, flash-frozen in liquid nitrogen, and then stored at –80 °C.

**In vitro HelD–RNAP complex reconstitution for cryo-EM**. To assemble the HelD–RNAP complex, the individual proteins were mixed at a molar ratio of 3:1. The in vitro reconstitutions were carried out at 4 °C, and the reconstitution mixture was incubated for 15 min. 50 μl of the reconstitution mixture was injected onto a Superose 6 Increase 3.2/300 column (GE Healthcare) equilibrated in 20 mM Tris–HCl, pH 7.8 (4 °C), 150 mM NaCl, 10 mM MgCl_2, and 1 mM DTT. 50-μl fractions were collected and the protein was eluted at ~1 μM concentration.

**Electron microscopy**. Complexes were diluted to ~850 nM and aliquots of 3 μl were applied to Quantifoil R1.2/1.3 or R2/2 Au 300 mesh grids, immediately blotted for 2 s, and plunged into liquid ethane using an FEI Vitrobot IV (4 °C, 100% humidity).

The grids were loaded into an FEI Titan Krios electron microscope at the European Synchrotron Radiation Facility (ESRF) (beamline CM01, ESRF, Grenoble) or CEITEC (Masaryk University, Brno), operated at an accelerating voltage of 300 keV and equipped with a post-GIF K2 Summit direct electron camera (Gatan) operated in counting mode. Cryo-EM data was acquired using EPU software (FEI) at a nominal magnification of ×165,000, with a pixel size of 0.8311 and 0.840 Å per pixel. Movies of a total fluence of ~40–50 electrons per Å² were collected at ~1 $e^-$/Å² per frame. A total number of 15,177 movies were acquired at a defocus range from −0.7 to −3.3 μm (Supplementary Table 6).

**Cryo-EM image processing.** All movie frames from three data sets were aligned and dose-weighted using the MotionCor2 program (Supplementary Fig. 3a) and then used for contrast transfer function parameter calculation with Gctf[38]. Initially, particles were selected without a template by Gautomatch (provided by Dr. Kai Zhang, http://www.mrc-lmb.cam.ac.uk/kzhang) from a small portion of the data set (~200 movies). This initial small dataset was subjected to reference-free 2D-classification using RELION 3.0[39]. Eight representative classes of different views were selected from the two-dimensional averages and used as a reference for automatic particle picking for the dataset I by RELION. WARP[40] was used for particle picking for data sets II and III.

The resulting particles were iteratively subjected to two rounds of 2D-classification (Supplementary Fig. 3b) at 3× and 2× binned pixel size. Particles in classes with poor structural features were removed. Particles from data sets I and II were globally refined to estimate the pixel size matching[41] and particles from dataset II were estimated to match the common pixel size 0.8311 Å per pixel. Particles from all data sets were pooled (~1560 k), 2× binned, and subjected to three-dimensional classifications with image alignment (Supplementary Fig. 4). The first round of 3D classification was restricted to ten classes and performed using *Msm* RNAP core (PDB ID 6F6W) as a 60 Å low-pass filtered initial model. The classification was done during three rounds of 25 iterations each, using regularization parameter $T = 4$. During the second and third round, local angular searches were performed at 3.5° and 1.8° to clearly separate structural species. The three most abundant and defined 3D classes were re-extracted at the pixel size of 0.8311 Å per pixel and 3D auto-refined using respective masks in RELION 3.0 (Supplementary Fig. 4). The results of the 3D auto-refinement were used for per-particle CTF refinement in RELION 3.1[42] and further 3D auto-refined. The further 3D classification was applied on classes 1 and 3 (corresponding to States I and III, respectively), but no better defined 3D classes were identified. The 3D reconstruction of class 2 (corresponding to State II) was further focused 3D auto-refined on the RNAP core region. The 3D reconstruction of class 2 was also 3D focus classified on the region of the HelD-specific domain and a more defined class was identified and 3D auto-refined separately. The final cryo-EM density maps were generated by the post-processing feature in RELION and sharpened or blurred into MTZ format using CCP-EM[43]. The resolutions of the cryo-EM density maps were estimated at the 0.143 gold standard Fourier Shell Correlation (FSC) cut off (Supplementary Fig. 3d). A local resolution (Supplementary Fig. 5a) was calculated using RELION and reference-based local amplitude scaling was performed by LocScale[44]. The directional resolution anisotropy (Supplementary Fig. 6) was quantified by the 3D FSC algorithm[45].

**Cryo-EM model building and refinement.** Atomic models of *Msm* RNAP protein parts (Fig. 1b–d) were generated according to the known structure of the *Msm* RNAP core (PDB entry 6F6W). The whole RNAP core was first rigid-body fitted into the cryo-EM density by Molrep[46], individual sub-domains were optimized using the Jigglefit tool[47] in Coot[48] and best fits were chosen according to a correlation coefficient in the JiggleFit tool. The crystal structure of the *Bsu* HelD-2A domain (Supplementary Fig. 9g) was first rigid-body fitted into the cryo-EM density by Molrep[46] and then manually adapted in Coot. Parts of the HelD main chain were first traced into the cryo-EM density by Buccaneer[49] and Mainmast[50]. The rest of the HelD protein was built *de-novo* in Coot[48]. The cryo-EM atomic models of HelD–RNAP complexes were then iteratively improved by manual building in Coot and refinement and validation with Phenix real-space refinement[51]. The atomic models were validated with the Phenix validation tool (Supplementary Table 6) and the model resolution was estimated at the 0.5 FSC cut off. Structures were analyzed and Figures were prepared using the following software packages: PyMOL (Schrödinger, Inc.) with APBS plugin[52], USCF Chimera[53], CCP4mg[54], and PDBePISA server[55].

**X-ray crystal structure determination of the *Bsu* HelD C-terminal domain.** The DNA sequence encoding the C-terminal domain of HelD (from residue 608 to 774) was amplified by PCR and cloned into pET15b vector by *Nde*I and *Bam*HI restriction sites to have an N-terminal His6-tagged protein. A bacterial culture containing BL21(DE3) RIPL codon-plus cells transformed with a pET15b–HelD-CTD vector was grown at 37 °C in LB medium supplemented with 100 μg/ml ampicillin, protein expression was induced with 0.5 mM IPTG at $OD_{600} = 0.5$, and incubated for additional 3 h to allow protein expression. Cells were harvested by centrifugation and lysed by sonication in lysis buffer (50 mM Tris–HCl, pH 8.0 at 4 °C, 200 mM NaCl, 5% glycerol, 2 mM β-mercaptoethanol, 2 mM phenylmethylsulfonyl fluoride, PMSF). The lysate was clarified by centrifugation and

HelD-CTD was purified by Ni-NTA, Q-sepharose, and Heparin-column chromatography. Fractions containing HelD-CTD were concentrated using VivaSpin concentrators until 10 mg/ml in crystallization buffer (10 mM Tris–HCl, pH 8 at 4 °C, 50 mM NaCl, 1% glycerol, 0.1 mM EDTA, 1 mM DTT).

Crystallization condition of HelD-CTD was screened by using the JCSG+ screen (Molecular Dimensions) and crystals were obtained in crystallization solution (0.1 M Na/K phosphate, pH 6.2, 0.2 M NaCl, 50% PEG200) at 22 °C. X-ray crystallographic data were collected at the Penn State X-ray Crystallography Facility and the data were processed with HKL2000[56]. For Sulfur single-wavelength anomalous dispersion phasing, 10 S atom positions were identified and the initial phase and density-modified map were calculated by AutoSol followed by automated model building by AutoBuild in the program Phenix[51]. Iterative refinement by Phenix and model building using Coot[48] improved the map and model. Finally, water molecules were added to the model. The data statistics and X-ray structure parameters are shown in Supplementary Table 7.

**Protein purification for biochemical assays—*Msm* RNAP core.** A strain of *Eco* containing plasmid with subunits of the RNAP core (LK1853[1]) was grown to the exponential phase ($OD_{600} \sim 0.5$). Expression of RNAP was induced with 500 μM IPTG for 4 h at room temperature. Cells were harvested by centrifugation, washed, resuspended in P buffer (300 mM NaCl, 50 mM $Na_2HPO_4$, 5% glycerol, 3 mM β-mercaptoethanol), and disrupted by sonication. Cell debris was removed by centrifugation and supernatant was mixed with 1 ml Ni-NTA Agarose (Qiagen) and incubated for 90 min at 4 °C with gentle shaking. Ni-NTA Agarose with bound RNAP was loaded on a Poly-Prep® Chromatography Column (BIO-RAD), washed with P buffer and, subsequently, washed with P buffer with 30 mM imidazole. The proteins were eluted with P buffer containing 400 mM imidazole and fractions containing RNAP were pooled and dialyzed against storage buffer (50 mM Tris–HCl, pH 8.0, 100 mM NaCl, 50% glycerol, 3 mM β-mercaptoethanol). The RNAP protein was stored at −20 °C.

**Protein purification for biochemical assays—*Msm* σ^A.** Expression strain of *Eco* containing plasmid with gene of σ^A (LK1740) was grown at 37 °C until $OD_{600}$ reached ~0.5; expression of σ^A was induced with 300 μM IPTG at room temperature for 3 h. Isolation of σ^A was done in the same way as RNAP purification with the exception of 50 mM imidazole added to the P buffer before resuspending the cells. Instead of the purification in a column, batch purification and centrifugation were used to separate the matrix and the eluate.

**Protein purification for biochemical assays—*Msm* RbpA.** The expression and purification of RbpA (LK1254, this work) were done in the same way as for RNAP except when $OD_{600}$ reached ~0.5, the expression was induced with 800 μM IPTG at room temperature for 3 h.

**Protein purification for biochemical assays—*Msm* HelD.** *Msm* HelD was prepared as described previously, in the paragraph about the purification of proteins for cryo-EM experiments.

The purity of all purified proteins was checked by SDS-PAGE gel.

**_Msm_ HelD ATP, GTP, and CTP hydrolysis assay.** Hydrolysis of ATP, GTP, and CTP (Sigma-Aldrich) by *Msm* HelD was measured in a total volume of 50 μl of reaction mixture which contained 10 mM substrate, 10 μg of *Msm* HelD, and reaction buffer composed out of 50 mM Tris–HCl, pH 7.5, 50 mM NaCl, 5 mM $MgCl_2$. Incubation was carried out at 37 °C for 30 min. The amount of released phosphate was analyzed spectrophotometrically at $\lambda = 850$ nm according to a modified molybdenum blue method[57] using a microplate reader Clariostar (BMG LAB-TECH, Ortenberg, Germany). Briefly, the reaction was stopped by adding 62 μl of reagent A (0.1 M L-ascorbic acid, 0.5 M $Cl_3CCOOH$). After thorough mixing, 12.5 μl of reagent B (10 mM $(NH_4)_6Mo_7O_{24}$) and 32 μl of reagent C (0.1 M sodium citrate, 0.2 M $NaAsO_2$, 10% acetic acid) was added. All enzymatic reactions were performed in triplicates with separate background readings for each condition.

**DNA–protein interaction analysis in vitro.** DNA–protein interactions were analyzed on 4–16% Bis-Tris native gels (ThermoFisher Scientific, cat. no. BN1002BOX) by Electrophoretic Mobility Shift Assay (EMSA). The DNA fragment was amplified by Expand High Fidelity PCR System (Roche, cat. no. 11732650001) using #1101 and #1146 primers and *Msm* chromosomal DNA. The resulting 304-bp-long PCR fragment was excised and purified from agarose gel. Binding reactions were performed in 1×STB buffer (50 mM Tris–HCl pH 8.0; 5 mM $Mg(C_2H_3O_2)_2$; 100 μM DTT; 50 mM KCl; 50 μg/ml BSA) that contained RNAP (25 pmol), HelD (125 pmol) and DNA (0.2 pmol). First, RNAP was pre-incubated in the presence or absence of HelD (at 37 °C, 45 min). Subsequently, DNA was added and samples were incubated at 37 °C for an additional 45 min. Then, NativePage buffer (Invitrogen, cat. no. BN2003) was added, and samples were loaded on a native gel. Electrophoresis was run in a cold room (4 °C). Finally, the gel was stained with DNA stain GelRed (Biotium, cat. no. 41003) in 1×TBS for 25 min and images were taken with an Ingenius UV-light camera (Syngen). Unbound DNA was quantified by the Quantity One software (BIO-RAD). The gel

was subsequently stained with Simply Blue (Invitrogen, cat. no. LC6060) for protein visualization.

**Protein–protein interaction analysis in vitro**. Protein–protein interactions were analyzed on 7% Tris-acetate native gels (ThermoFisher Scientific, cat. no. EA0355BOX) by EMSA. The binding reaction was done in 20 µl of 1×STB buffer containing RNAP (25 pmol), HelD (125 pmol), σ$^A$ (1250 pmol), and RbpA (1250 pmol)—protein combinations in reactions are specified in the Fig. 4 legend. First, RNAP was reconstituted with/without HelD (at 37 °C, 45 min). Then RbpA and/or σ$^A$ were added, followed by additional incubation at 37 °C for 45 min. 20 µl of Native Tris-Glycine buffer (Invitrogen, cat. no. LC2673) was added, and 20 µl of the mixture was then loaded on a native gel. Electrophoresis was run in a cold room (4 °C). Subsequently, for protein visualization, the gels were stained with Simply Blue. The identity of proteins in each band was determined by MALDI mass spectrometric identification.

**Disassembly of elongation complexes**. Elongation complexes (ECs), containing a transcription bubble, were assembled with the *Msm* RNAP core, based on a previously described assay[58]. Briefly, DNA and RNA oligonucleotides were purchased and are the same as in Table EV7 in[59]. The RNA (LK-pRNA) was monophosphorylated at the 5′ end by the manufacturer. A 2-fold molar excess of RNA was mixed with template DNA (LK632) in water and annealed in a cycler (45 °C for 2 min, 42–27 °C: the temperature was decreasing by 3 °C every 2 min, 25 °C for 10 min). RNAP (32 pmol per sample) was incubated with 4 pmol of the annealed hybrid in 10 µl of reaction buffer (40 mM Tris–HCl, pH 8.0, 10 mM MgCl$_2$, 1 mM DTT) for 15 min at room temperature with gentle shaking. 8 pmol of non-template DNA (LK631) containing biotin at the 5′ end was added and the mixture was incubated at 37 °C for 10 min.

Streptavidin-coated magnetic beads (25 µl per sample; Sigma S-2415) were washed with 500 µl of binding buffer (20 mM Tris–HCl, pH 8.0, 0.15 M NaCl) and resuspended in the same volume of fresh binding buffer. Assembled elongation complexes were then mixed with washed beads. ECs and beads were incubated together for 30 min at RT (room temperature) with continuous gentle shaking. Unbound complexes were removed by subsequent washing with 500 µl of binding buffer, 500 µl of washing buffer (20 mM Tris–HCl pH 8.0, 0.5 M NaCl, 2 mM MgCl$_2$, 1 mM DTT) and 500 µl of reaction buffer[60]. Beads were resuspended in reaction buffer with 100 mM final concentration of KCl, with or without GTP or ATP (final concentration 200 µM) in a total volume of 5 µl. HelD in the 2-fold ratio over RNAP (64 pmol per sample) or heat-inactivated HelD (5 min at 95 °C) or buffer were added to the final reaction volume of 10 µl. Reactions proceeded for 20 min at 37 °C. The bound (in complex with EC) and released (free in buffer) RNAPs were separated by using a DYNAL Invitrogen bead separation device. Subsequently, the fractions containing released RNAPs were spotted directly on the nitrocellulose membrane. RNAPs were detected by western blotting using mouse monoclonal antibodies against the β subunit of RNAP (clone name 8RB13, dilution 1:1000) and secondary antibodies conjugated with a fluorophore dye (WesternBright™ MCF-IR, Advansta, 800 nm anti-mouse antibody, dilution 1:10 000) and scanned with an Odyssey reader (LI-COR Biosciences). The analysis was done with the Quantity One software (BIO-RAD). The experiment was conducted in five biological replicates.

**Immunoprecipitation**. 150 ml of *Msm* exponential (Supplementary Fig. 12) and 100 ml of stationary phase (Fig. 4f and Supplementary Figs. 1 and 13) cells were pelleted and resuspended in 4 ml of Lysis buffer (20 mM Tris–HCl, pH 8, 150 mM KCl, 1 mM MgCl$_2$) with 1 mM DTT, 0.5 mM PMSF and Sigma protease inhibitor cocktail P8849 (5 µl/ml), sonicated 15 × 10 s with 1 min pauses on ice and centrifuged. 1 ml of stationary and 1.5 ml of exponential phase cells lysates were incubated overnight at 4 °C with 25 µl of ANTI-FLAG® M2 Affinity Agarose Gel (Sigma, A2220). Agarose gel beads with the captured protein complexes were washed 4× with 0.5 ml of lysis buffer. FLAG-tagged proteins were eluted by 60 µl of 3× FLAG® Peptide (Sigma F4799) (diluted in Tris-buffered saline (TBS) to a final concentration of 150 ng/ml). Proteins were resolved on sodium dodecylsulphate-polyacrylamide gel electrophoresis (SDS-PAGE) and Simply Blue-stained (SimplyBlue, Invitrogen) or analyzed by western blotting.

**Double pull-down**. Eluted proteins from the first immunoprecipitation (ANTI-FLAG, see above) from lysates of the HelD-FLAG culture from exponential phase were incubated (O/N, 4 °C) with 5 µg of σ$^A$ or IgG antibodies (negative control), respectively, bound to 20 µl of Protein G-plus Agarose (Santa Cruz Biotechnology, cat. no. sc-2002), and then 4× washed with 1 ml Lysis buffer. Finally, proteins were analyzed by SDS-PAGE and western blot.

**Western blotting**. Proteins were resolved by SDS-PAGE and detected by western blotting using mouse monoclonal antibodies against σ$^{70}$/σ$^A$ (clone name 2G10, Biolegend, cat. no. 663208, dilution 1:1000), against the β-subunit of RNAP (clone name 8RB13, Biolegend, cat. no. 663903, dilution 1:1000), monoclonal anti-FLAG (clone M2, Sigma cat. no. F1804, dilution 1:1000), and anti-mouse secondary antibodies conjugated with HRP (Sigma, cat. no. A7058, dilution 1:80 000). Subsequently, the blot was incubated for 5 min with SuperSignal™ West Pico PLUS Chemiluminiscent substrate (ThermoScientific, cat. no. 34577), exposed on film, and developed.

**Trypsin digestion and MALDI mass spectrometric identification**. Simply Blue-stained protein bands were cut out from gels, chopped into small pieces, and destained using 50 mM 4-ethylmorpholine acetate (pH 8.1) in 50% acetonitrile (MeCN). The gel pieces were then washed with water, reduced in size by dehydration in MeCN, and partly dried in a SpeedVac concentrator. The proteins were digested overnight at 37 °C using sequencing grade trypsin (100 ng; Promega) in a buffer containing 25 mM 4-ethylmorpholine acetate and 5% MeCN. The resulting peptides were extracted with 40% MeCN/0.2% TFA (trifluoroacetic acid).

For MALDI MS analysis, 0.5 µl of each peptide mixture was deposited on the MALDI plate, air-dried at room temperature, and overlaid with 0.5 µl of the matrix solution (α-cyano-4-hydroxycinnamic acid in 50% acetonitrile/0.1% TFA; 5 mg/ml, Sigma). Peptide mass maps of proteins in Fig. 4f and Supplementary Fig. 1 were measured using an Autoflex Speed MALDI-TOF instrument (Bruker Daltonics, Billerica, USA) in a mass range of 700–4000 Da and calibrated externally using a PepMix II standard (Bruker Daltonics). For protein identification, MS spectra were searched against NCBIprot_20190611 database subset of bacterial proteins using the in-house MASCOT search engine v.2.6 with the following settings: peptide tolerance of 20 ppm, missed cleavage site set to one, and variable oxidation of methionine. The spectra of proteins in Fig. 4h were acquired on a 15T Solarix XR FT-ICR mass spectrometer (Bruker Daltonics) in a mass range of 500–6000 Da and calibrated internally using peptide masses of *Msm* RpoB and RpoC proteins. The peak lists generated using DataAnalysis 5.0 program were searched against UniProtKB database of *Msm* proteins using the in-house MASCOT engine with the following settings: peptide tolerance of 3 ppm, missed cleavage site set to two variable, and oxidation of methionine.

**Reporting summary**. Further information on research design is available in the Nature Research Reporting Summary linked to this article.

## Data availability

Data are available from the corresponding authors. Coordinates and structure factors or maps have been deposited in the wwwPDB or EMDB: *Bsu* HelD C-terminal domain (X-ray) PDB ID 6VSX, *Msm* HelD–RNAP complex State I (cryo-EM) EMD-10996, PDB ID 6YXU, *Msm* HelD–RNAP complex State II (cryo-EM) EMD-11004, PDB ID 6YYS, *Msm* HelD–RNAP complex State III (cryo-EM) EMD-11026, PDB ID 6Z11. Source data are provided with this paper.

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

## Acknowledgements

We thank the ESRF (especially Michael Hons), IBS, and EMBL for access to the ESRF Krios beamline CM01; the CEITEC and the Czech Infrastructure for Integrative Structural Biology (CIISB) for access to the CEITEC Krios microscope and to the CMS facilities at BIOCEV (project LM2015043 by MEYS). This work was supported by 20-12109S (to L.K. and J.Do.) and 20-07473S (to J.H.) from the Czech Science Foundation, NIH grant R35 GM131860 to K.M., and by the Academy of Sciences of the Czech Republic (RVO: 86652036), MEYS (CZ.1.05/1.1.00/02.0109), European Regional Development Fund (Project CIISB4HEALTH, No. CZ.02.1.01/0.0/0.0/16_013/0001776 and ELIBIO, No. CZ.02.1.01/0.0/0.0/15_003/0000447). T.Kou. holds a fellowship from the EMBL Interdisciplinary Postdocs (EI3POD) initiative co-funded by Marie Skłodowska-Curie grant agreement no. 664726.

## Author contributions

J.Do. and L.K. conceived and supervised the project. T.Kou., T.Kov., M.T., and J.Du. expressed and purified proteins for cryo-EM, TKou prepared cryo-EM grids, collected cryo-EM data together with J.N., performed image processing and 3D reconstruction, and built initial models together with J.Do. K.S.M. and U.C. solved the *Bsu* HelD CTD. M.J., J.H., B.B., M.Š., J.P., P.S., and H.Š. did cloning, protein purifications, and IPs. P.H. identified proteins by mass spectrometry. J.P. and P.S. performed DNA binding experiments. M.T., J.Du., and T.Kov. performed NTP hydrolysis experiments. T.Kou., T. Kov., and J.Do. built and refined atomic models and created figures. I.B. and M.S. performed in silico modeling. T.Kou., T.Kov., J.Do., and L.K. wrote the manuscript with input from I.B. T.S. performed initial modeling and comparative analysis.

## Competing interests

The authors declare no competing interests.
