## [Peer Review File · Nature Communications]

REVIEWER COMMENTS

Reviewer #1 (Remarks to the Author):

This manuscript reports high-resolution cryo-EM structures of a mycobacteria RNA polymerase in complex with an RNAP-recycling factor, HelD. The three complexes, obtained at resolutions of 3.1, 3.1, and 3.5 angstroms, respectively, reveal different states of HelD-RNAP engagement and provide structural details of HelD-induced nucleic-acid clearance and RNAP recycling. The transcription termination and post-termination stages are the least understood stage of transcription. This beautiful structure work elucidates an unexpected delicate mechanism of transcription regulation occurring at transcription termination and will be of clear interest to researchers in bacterial transcription and bacterial transcriptional regulation. The manuscript should be acceptable for publication after a very minor revision as follows:

1. Include a scale bar for Fig. S3c.
2. Line 110 "primary and secondary channels"
3. Line 204, "in RapA'.
4. Fig. 2e, why a negative value for CTP was obtained?
5. Page 18 and 19 in the supplemental information are reversed.
6. Fig S12, label HelD residues making contacts with RNAP, and discuss whether the interface residues are conserved.
7. Line 268, do the four residues (482-DEED-485) make coordinate bonds with MgA? In Fig 3C, the coordinate bonds with MgA made by the AS aspartate triad and HelD residues (if any) should be shown.

Reviewer #2 (Remarks to the Author):

Kouba et al. have presented a fascinating, new, and significant structure. The HelD factor binds RNAP in an unprecedented manner that gives insights into its mechanism. HelD appears to pry open the clamp, and functional data here support that that it leads to the release of nonspecific DNA and stalled elongation complexes. What's not clear its role is in initiation, and I'm perplexed why the authors go through so much effort to show binding to the initiation factors and yet no functional assays. I recommend that section either be removed until the significance is addressed or the authors expound on the experiments required to address this. It's disappointing no functional initiation studies were done. That said, the manuscript brims with new data and analyses. However, some issues need to be fixed, and I have some suggestions to make the manuscript easier to read. I recommend accept with revision. One major required revision is noted below.

The way the particles were classified is not satisfactorily explained and in my opinion, orthodox. The authors basically combined the original class II with a subset of class II – in other words, they classified for a HelD class then recombined with particles that may not have HelD. It is also not clear if this is a composite map. This is a highly flawed approach as they are combining RNAP particles without HelD to that with HelD and claiming changes in RNAP. I understand they are trying to get higher resolution, but it's not really as the majority of particles don't have that HelD. They should show what the other class was (the low res class- is it junk?- this is important given they merge the subclass back with this "low res class"). Also, why did they ask for only two classes, usually on focused classification you ask for several classes to sort out heterogeneity and address why only do focused classification on state II? They would then need to submit the new maps as I suspect the resolution will drop. I don't think it will change their conclusion, but I would need to see.

Here are my suggestions to strengthen the manuscript:

The significant observations are buried with the details. It's difficult to understand the significance of all the interactions without context. I suggest the authors define all the RNAP modules they will discuss and describe their roles in transcription. In the intro, they need to introduce all these parts of RNAP and their roles and then mention their structures make interactions/affects these modules. Those unacquainted for RNAP will need a sentence or two explaining the significance of the rudder, b-lobe, shelf, jaw etc. before delving into these results. There needs to be figure with the overall structure highlighting these features. Then next to that figure one with HeID bound. Something similar to figure one but with the TL, rudder also (a sliced view).

It is also not clear the exact transition between the "states"; the authors could use 3D-variability – however, I leave that at their discretion.

82-87: See above: need to be rewritten to match fig. S4 better. What is the subclass at 3.6? State II is actually that subclass combined with the original state II? The way this map was made is very confusing and unconventional as it is a combination of the original particles in state II and a subset of particles- combined? Then the authors still call it State II (so there are actually two state IIs). This data needs to be reprocessed. They should classify particles with HeID from state II, they can then locally refine around HeID and make a composite map. The way state II is processed is not correctly done (basically combining a sub class with the original class to "get" higher overall resolution) and I fear might misrepresent the conformation of the RNAPS as most of the particles composing that part of the map comes from RNAPS not bound to HeID.

Lines 34-37 need to be written more clearly. What do they mean it "dissociates transcription elongation complexes?" – does it remove RNAP from the DNA during elongation? Also, break into two sentences regarding sigma factor, and those two activities are not related.

Line 40- what are "undesirable nucleic acids"? Please be more specific.

Line 50-CarD is widespread not only found in actinobacteria

Line 51- I would argue that the factors help regulate and coordinate RNAP function– rather than use the term "smooth."

58-59: weird phrasing. Break into two sentences.

66: change to "upon addition of ATP."

Figures:

It would be better to show S4 before S3. To describe the quality of the classes before explaining how the classes were attained is a little awkward.

Fig. S3 FG-are these maps carved? If so, please list the carving cutoff as carving around a pdb with too tight of a carve can create the illusion of higher res.

Fig. 3D- the authors need to label what parts of the structure are shown by local resolution. To do so, they can have the pdb placed directly next to these local resolution maps and highlight HeID or draw an outline of HeID on the local res maps. A general criticism is that an overall density map is not shown for the fitted pdb and given that some of HeID was built de novo that is concerning. The readers should not have to download the maps to determine if the map warrants the structure. A density mesh map around HeID (uncarved) would be ideal with the local resolution map next to it.

Fig. 3, can they give a reference view of the overall structure.

272: the descriptor "only" is strange and not scientific. Do they mean less than something else?

277: dwDNA? Please write out abbreviations first time used

286-290: This statement is confusing. Can the authors say why? Also, the word "probably" is strange here- maybe say what about their findings suggest this. What's the basis of this proposal?

295: what are core2 and holo2? I'm familiar with this group's work, but many readers might not know. Why core2 and holo2 and not core1 and holo1? Can they give some context?

305-306: This statement is not correct. All of the states would release the DNA as the clamp is wide open. Opening or rotating an already widened clamp that can't bind DNA is not what precipitates or facilitates DNA release.

Lines 403: change "expulsion" to "release"- expulsion suggests an active process.

If the authors chose to include the co-IPs with sigma and RbpA, they should do some functional experiments. Does HeID inhibit transcription? This is easy enough to test. Alternatively, because the manuscript is already rich with data, they could perhaps propose what definitive experiment should be performed

392: needs to state that Figure 5 is a model. State in the title of the legend too.

5g- does not add to our understanding and no data is addressing this other than HeID binds when sigma and RbpA binds but this schematic does not show this. Remove 5g. The rest of the figure is very clear and informative.

Reviewer #1 (Remarks to the Author):

This manuscript reports high-resolution cryo-EM structures of a mycobacteria RNA polymerase in complex with an RNAP-recycling factor, HeID. The three complexes, obtained at resolutions of 3.1, 3.1, and 3.5 angstroms, respectively, reveal different states of HeID-RNAP engagement and provide structural details of HeID-induced nucleic-acid clearance and RNAP recycling. The transcription termination and post-termination stages are the least understood stage of transcription. This beautiful structure work elucidates an unexpected delicate mechanism of transcription regulation occurring at transcription termination and will be of clear interest to researchers in bacterial transcription and bacterial transcriptional regulation. The manuscript should be acceptable for publication after a very minor revision as follows:

1. Include a scale bar for Fig. S3c.

RESPONSE

a/ The analysis from Cryo EM does not provide any scale for the angular distribution of particle projections. Each projection is visualized with a single blue spot in the Mollweide projection. The more covered areas thus appear as darker blue areas in the visualization.

b/ Possibly, Fig. 3S ab could be meant.

ACTION TAKEN

a/ We included a 3D FSC analysis (Supplementary Figure 6) that illustrates that the sample does not suffer from the so called 'preferred orientation' problem. We also included the efficiency score and sphericity, which is shown in Supplementary Table 1.

b/ We added scale bars to Supplementary Figure 3ab.

2. Line 110 "primary and secondary channels"

ACTION TAKEN

Done.

3. Line 204, "in RapA'.

ACTION TAKEN

Done.

4. Fig. 2e, why a negative value for CTP was obtained?

RESPONSE

The apparent negative value of CTP hydrolysis is probably caused by the intrinsic contamination of CTP with free phosphate ions. These free ions are the source of a relatively high background (when the colorimetric method is used) and this caused the CTP value to be negative. To conclude, it means

that the value is effectively zero but we felt that reporting it in the original form was the best approach.

ACTION TAKEN

We show the individual data points, as requested, and explanation is added to the Fig. 2e legend.

5. Page 18 and 19 in the supplemental information are reversed.

RESPONSE

Thank you! Our mistake.

ACTION TAKEN

Corrected.

6. Fig S12, label HelD residues making contacts with RNAP, and discuss whether the interface residues are conserved.

RESPONSE

Thank you for this suggestion. We are planning to write a mini-review describing the recent structural findings about the HelD proteins from *M. smegmatis* and *B. subtilis* where we will discuss the conservation and functional relevance of HelD-RNAP contacts in detail.

ACTION TAKEN

In this manuscript, in Supplementary Figure 14, we marked the contacts (amino acids) of *Msm* HelD that interact with the RNAP core as observed in State II (Table S1) with green rectangles.

7. Line 268, do the four residues (482-DDED-485) make coordinate bonds with MgA? In Fig 3C, the coordinate bonds with MgA made by the AS aspartate triad and HelD residues (if any) should be shown.

RESPONSE

Only Asp483 of the HelD PCh-loop 482-DDED-485 of State II coordinates the MgA cation. Asp482 is only in close proximity but not coordinating MgA; the remaining acids are too distant to interact in the reported structures.

ACTION TAKEN

We added the coordination bonds of the Asp triad to Fig. 3f (originally 3c) together with one coordination bond of Asp483 of the HelD PCh-loop.

Reviewer #2 (Remarks to the Author):

Kouba et al. have presented a fascinating, new, and significant structure. The HelD factor binds RNAP in an unprecedented manner that gives insights into its mechanism. HelD appears to pry open the clamp, and functional data here support that that it leads to the release of nonspecific DNA and

stalled elongation complexes. What's not clear its role is in initiation, and I'm perplexed why the authors go through so much effort to show binding to the initiation factors and yet no functional assays. I recommend that section either be removed until the significance is addressed or the authors expound on the experiments required to address this. It's disappointing no functional initiation studies were done. That said, the manuscript brims with new data and analyses. However, some issues need to be fixed, and I have some suggestions to make the manuscript easier to read. I recommend accept with revision. One major required revision is noted below.

The way the particles were classified is not satisfactorily explained and in my opinion, orthodox. The authors basically combined the original class II with a subset of class II – in other words, they classified for a HelD class then recombined with particles that may not have HelD. It is also not clear if this is a composite map. This is a highly flawed approach as they are combining RNAP particles without HelD to that with HelD and claiming changes in RNAP. I understand they are trying to get higher resolution, but it's not really as the majority of particles don't have that HelD. They should show what the other class was (the low res class- is it junk?- this is important given they merge the subclass back with this "low res class"). Also, why did they ask for only two classes, usually on focused classification you ask for several classes to sort out heterogeneity and address why only do focused classification on state II? They would then need to submit the new maps as I suspect the resolution will drop. I don't think it will change their conclusion, but I would need to see.

RESPONSE

We have clarified the classification process of State II in Figure S4. Please note that we did not combine any HelD bound and unbound classes of particles, all State II particles contained bound HelD. The deposited EMD-11004 map is focus-refined around the region of the RNAP core, the HelD N-terminal domain, the 1A domain, and the PCh-loop. The filtering at the estimated overall resolution (3.1 Å) scatters (due to over-sharpening) the density for the rest of the HelD protein and the β'-clamp but these regions are still present in blurred maps (at a resolution range of ~4-6 Å; please, see the State II local res. Map in Supplementary Figure 5a).

Please see more details regarding the HelD-specific domain sub-classification in our response to your specific comment for lines 82-87.

We hope that we have clarified that we did not use any flawed approach and that our maps are a true representation of the observed Msm HelD-RNAP complexes (see also ACTION TAKEN).

ACTION TAKEN

To avoid a misunderstanding, we deposited the LocScale map, which enables local filtering and better representation of the overall cryo-EM map.

To Supplementary Figure 4 we also added all the low-resolution classes from the initial 3D classification.

We indeed tried to sub-classify particles from states I and III, but the resultant classes did not give rise to any more defined classes, and only generated reconstructions with lower overall resolution. We mentioned that in the Methods section (starting on line 716).

Here are my suggestions to strengthen the manuscript:

The significant observations are buried with the details. It's difficult to understand the significance of all the interactions without context. I suggest the authors define all the RNAP modules they will

discuss and describe their roles in transcription. In the intro, they need to introduce all these parts of RNAP and their roles and then mention their structures make interactions/affects these modules. Those unacquainted for RNAP will need a sentence or two explaining the significance of the rudder, b-lobe, shelf, jaw etc. before delving into these results. There needs to be figure with the overall structure highlighting these features. Then next to that figure one with HeID bound. Something similar to figure one but with the TL, rudder also (a sliced view).

RESPONSE

We agree.

ACTION TAKEN

We inserted a paragraph to the Introduction, describing the topology of RNAP, and the functions of the structural elements described (starting on line 48). We also added panels to Figures 1 and 3 (1a and 3a) showing the *Msm* RNAP core with these structural elements marked. Also, we marked the respective topological features in Supplementary Movies 1-3.

It is also not clear the exact transition between the “states”; the authors could use 3D-variability – however, I leave that at their discretion.

RESPONSE

We appreciate this suggestion; we might use this type of analysis in our future work.

82-87: See above: need to be rewritten to match fig. S4 better. What is the subclass at 3.6? State II is actually that subclass combined with the original state II? The way this map was made is very confusing and unconventional as it is a combination of the original particles in state II and a subset of particles- combined? Then the authors still call it State II (so there are actually two state IIs). This data needs to be reprocessed. They should classify particles with HeID from state II, they can then locally refine around HeID and make a composite map. The way state II is processed is not correctly done (basically combining a sub class with the original class to “get” higher overall resolution) and I fear might misrepresent the conformation of the RNAPS as most of the particles composing that part of the map comes from RNAPS not bound to HeID.

RESPONSE

Please, see also our RESPONSE to your first comment.

ACTION TAKEN

We clarified the data classification process of State II in Figure S4 so that it matches the description on lines 82-87 (now 99-104 and also, please, see text starting on line 716 in Mat&Met). Please note that all particles from State II always contain bound HeID. In order to get a better map for the HeID 1A and HeID-specific domains, we further classified the original set of State II particles on this specific region. This produced the subclass at 3.6 Å resolution, and three low resolution classes (now visualized in the processing scheme). We used the 3.6 Å resolution map as a parallel guideline for de-novo building of HeID-specific domains. We did not deposit this map, because the regions of the RNAP core are of low quality, hence not optimal for global atomic model refinement of the entire complex.

Lines 34-37 need to be written more clearly. What do they mean it “dissociates transcription

elongation complexes?” – does it remove RNAP from the DNA during elongation? Also, break into two sentences regarding sigma factor, and those two activities are not related.

RESPONSE

We agree. HelD dissociates stalled (non-functional) transcription complexes. In this way it “sweeps” the DNA and prevents e. g. transcription-replication collisions.

ACTION TAKEN

We changed the wording to: “We show that HelD prevents non-specific interactions between RNAP and DNA and dissociates stalled transcription elongation complexes.” Furthermore, we omitted the note on sigma factor in the Abstract to make the text flow better.

Line 40- what are “undesirable nucleic acids”? Please be more specific.

ACTION TAKEN

We changed the phrasing to “...releases RNAP from nonfunctional complexes...”.

Line 50-CarD is widespread not only found in actinobacteria.

RESPONSE

We agree.

ACTION TAKEN

As CarD does not feature further on in the manuscript, we deleted this mention.

Line 51- I would argue that the factors help regulate and coordinate RNAP function– rather than use the term “smooth.

RESPONSE

We agree.

ACTION TAKEN

We changed the phrasing to “...the regulation of the transcription machinery depends on concerted activities of RNAP and numerous transcription factors...”.

58-59: weird phrasing. Break into two sentences.

RESPONSE

We agree.

ACTION TAKEN

We broke the composite sentence into two sentences (lines 70-72).

66: change to “upon addition of ATP.

RESPONSE

This is a misunderstanding. The sentence does not mean “upon [i. e. after] addition of ATP”. The meaning is that ATP can be hydrolyzed by HeID, and also GTP can be hydrolyzed by HeID (independently of each other).

ACTION TAKEN

We changed the phrasing to: “...that in addition to being able to hydrolyze ATP, HeID can also hydrolyze GTP” (page 4, line 81).

Figures:

It would be better to show S4 before S3. To describe the quality of the classes before explaining how the classes were attained is a little awkward.

RESPONSE

We agree.

ACTION TAKEN

We created a new figure - S5. It contains a part of the information that was formerly in S3. Figure S5 shows the overall and local agreement of the cryo-EM maps and the final models, and better illustrates the logical flow of the cryo-EM data processing.

Fig. S3 FG-are these maps carved? If so, please list the carving cutoff as carving around a pdb with too tight of a carve can create the illusion of higher res.

RESPONSE

Fig. S5cd (formerly Fig. S3fg) are LocScale filtered maps visualized in the CCP4mg software package with a 1.75 Å clip radius.

ACTION TAKEN

We added the additional information to the Figure S5cd legend. We believe the LocScale filtering, which usually slightly blurs the original maps, gives an unbiased representation of the original (usually locally over-sharpened) maps.

Fig. 3D- the authors need to label what parts of the structure are shown by local resolution. To do so, they can have the pdb placed directly next to these local resolution maps and highlight HeID or draw an outline of HeID on the local res maps. A general criticism is that an overall density map is not shown for the fitted pdb and given that some of HeID was built de novo that is concerning. The readers should not have to download the maps to determine if the map warrants the structure. A density mesh map around HeID (uncarved) would be ideal with the local resolution map next to it.

RESPONSE

We agree.

ACTION TAKEN

In Fig. S5a (formerly Fig. S3d) we now show a cylinder model of all HeID-RNAP complexes in similar orientations as the local resolution maps. For States I and II we added a black line delineating the

HelD protein. We also added Fig. S5b showing a LocScale map for the HelD part of the HelD-RNAP complexes.

Fig. 3, can they give a reference view of the overall structure.

RESPONSE

We agree.

ACTION TAKEN

We added an overall view of the *Msm* RNAP core together with State I and State II as Fig. 3abc in a similar orientation as in Fig. 3de.

272: the descriptor “only” is strange and not scientific. Do they mean less than something else?

RESPONSE

The original idea (for using “only”) was to emphasize that even though the two tips of HelD belong to two distant arms of this protein, and even though the two tips interact with two different channels of RNAP, they come close together in State II of the complex.

ACTION TAKEN

The wording was changed to “the shortest distance between the two tips is about 17 Å”.

277: dwDNA? Please write out abbreviations first time used

RESPONSE

Done.

286-290: This statement is confusing. Can the authors say why? Also, the word “probably” is strange here- maybe say what about their findings suggest this. What’s the basis of this proposal?

RESPONSE

The statement was meant to explain that already the interaction of the HelD N-terminal domain with the secondary channel of RNAP is likely sufficient to cause some changes leading to a weakening of the RNAP-dwDNA interaction.

ACTION TAKEN

We changed the wording of the sentence to increase clarity (starting on line 323).

295: what are core2 and holo2? I’m familiar with this group’s work, but many readers might not know. Why core2 and holo2 and not core1 and holo1? Can they give some context?

ACTION TAKEN

We now refer more precisely (paragraph starting on line 328) to our previous work where we described the conformation of the primary channel opening of the *Msm* RNAP core and holoenzyme where we identified two conformations for both the RNAP core and holoenzyme and termed them Core1 and Core2, and Holo1 and Holo2 (please, see Table 1 in DOI: 10.1128/JB.00583-18). Based on the span of the primary channel in State III (~33 Å), the most similar structures regarding the primary channel opening correspond to Core2 and Holo2 (32.2 and 33.6 Å, respectively).

305-306: This statement is not correct. All of the states would release the DNA as the clamp is wide open. Opening or rotating an already widened clamp that can't bind DNA is not what precipitates or facilitates DNA release.

ACTION TAKEN

The phrasing was changed to better express the changes leading to DNA release (line 344).

Lines 403: change “expulsion” to “release”- expulsion suggests an active process.

ACTION TAKEN

Done (line 465).

If the authors chose to include the co-IPs with sigma and RbpA, they should do some functional experiments. Does HeID inhibit transcription? This is easy enough to test. Alternatively, because the manuscript is already rich with data, they could perhaps propose what definitive experiment should be performed

RESPONSE

In *B. subtilis*, HeID binds to the RNAP core, and previously published pull-down experiments revealed that *B. subtilis* SigA was not present in the complex (Wiedermannova et al., Nucl Acids Res 2014). In *M. smegmatis*, however, both SigA and also RbpA can be part of the complex as we demonstrated by the IP experiments in this manuscript. This is a qualitatively new finding, revealing another difference between the HeID proteins and the overall architectures of the transcription machineries in *B. subtilis* and *M. smegmatis*. We believe that even without additional functional tests, this piece of information complements the structural insights described in the manuscript. This finding will pave the way to our understanding of how HeID is released from RNAP, and experiments addressing this issue are already under way, and will be reported in due course.

ACTION TAKEN

We reorganized the text so that this information better fits into the context (starting with line 383).

392: needs to state that Figure 5 is a model. State in the title of the legend too.

ACTION TAKEN

Done.

5g- does not add to our understanding and no data is addressing this other than HeID binds when

sigma and RbpA binds but this schematic does not show this. Remove 5g. The rest of the figure is very clear and informative.

ACTION TAKEN

We removed 5g from figure 5.

REVIEWERS' COMMENTS

Reviewer #2 (Remarks to the Author):

The authors have done an excellent job addressing my initial review. The analysis is scholarly and thorough and support the structures. The manuscript is in great form and I recommend publishing. I have no specific comments other than to commend the authors on a fascinating finding and compelling model.